# Previous Usutu Virus Exposure Partially Protects Magpies (*Pica pica*) against West Nile Virus Disease But Does Not Prevent Horizontal Transmission

**DOI:** 10.3390/v13071409

**Published:** 2021-07-20

**Authors:** Estela Escribano-Romero, Nereida Jiménez de Oya, María-Cruz Camacho, Ana-Belén Blázquez, Miguel A. Martín-Acebes, Maria A. Risalde, Laura Muriel, Juan-Carlos Saiz, Ursula Höfle

**Affiliations:** 1Department of Biotechnology, Instituto Nacional de Investigación y Tecnología Agraria y Alimentaria (INIA-CSIC), 28040 Madrid, Spain; eescribano@inia.es (E.E.-R.); jdeoya@inia.es (N.J.d.O.); blazquez@inia.es (A.-B.B.); martin.mangel@inia.es (M.A.M.-A.); jcsaiz@inia.es (J.-C.S.); 2Health and Biotechnology Research Group SaBio, National Institute for Game Rearch IREC (University of Castilla—La Mancha UCLM-National Research Council CSIC-Regional Government of Castilla—La Mancha JCCM), 13005 Ciudad Real, Spain; mrcruzcamacho@gmail.com; 3Departamento de Anatomía y Anatomía Patológica Comparadas, Facultad de Veterinaria, Agrifood Excellence International Campus (ceiA3), Universidad de Córdoba (UCO), 14014 Córdoba, Spain; maria.risalde@uco.es (M.A.R.); lauramurielcantarero@hotmail.com (L.M.); 4Infectious Diseases Unit, Hospital Universitario Reina Sofía de Córdoba, Instituto Maimonides de Investigación Biomédica de Córdoba (IMIBIC), University of Córdoba, 14004 Córdoba, Spain

**Keywords:** avian host, *Flaviviruses*, co-infection, cross-protection, Usutu virus, West Nile virus, magpie

## Abstract

The mosquito-borne flaviviruses USUV and WNV are known to co-circulate in large parts of Europe. Both are a public health concern, and USUV has been the cause of epizootics in both wild and domestic birds, and neurological cases in humans in Europe. Here, we explore the susceptibility of magpies to experimental USUV infection, and how previous exposure to USUV would affect infection with WNV. None of the magpies exposed to USUV showed clinical signs, viremia, or detectable neutralizing antibodies. After challenge with a neurovirulent WNV strain, neither viremia, viral titer of WNV in vascular feathers, nor neutralizing antibody titers of previously USUV-exposed magpies differed significantly with respect to magpies that had not previously been exposed to USUV. However, 75% (6/8) of the USUV-exposed birds survived, while only 22.2% (2/9) of those not previously exposed to USUV survived. WNV antigen labeling by immunohistochemistry in tissues was less evident and more restricted in magpies exposed to USUV prior to challenge with WNV. Our data indicate that previous exposure to USUV partially protects magpies against a lethal challenge with WNV, while it does not prevent viremia and direct transmission, although the mechanism is unclear. These results are relevant for flavivirus ecology and contention.

## 1. Introduction

Usutu virus (USUV) is a mosquito-transmitted flavivirus that belongs to the Japanese encephalitis virus serocomplex [1]. After its first identification in South Africa in 1959 [2], the virus was detected in Europe in 2001 in an outbreak in birds in Austria [3], although retrospective identification in tissues from blackbird mortalities in Italy suggests it has been circulating in Europe since at least 1996 [4]. The virus has now spread throughout Europe, causing high bird mortality [5] and some human cases [6,7,8,9,10]. USUV seropositivity has been reported in more than 58 bird species belonging to 26 families and 13 orders, Passeriformes being the most affected [5,11]. In 2016, several countries in Western Europe reported the largest epizootic of the USUV registered so far in Europe, which caused a massive mortality of birds [5]. However, scarce experimental data on USUV infection in avian species are available [12,13,14].

At least two other mosquito borne-flaviviruses co-circulate with USUV in different regions of Europe. While the Bagaza virus (BAGV) has so far only affected birds [15], West Nile virus (WNV) is highly neurovirulent in humans [5,16]. During the last ten years, WNV outbreaks among birds and horses, and a worrying increase in human cases, with up to 2671 confirmed infections and 213 deaths, have been reported in the continent [17].

The magpie, one of the most abundant corvids in Europe [18], has recently been shown to be highly susceptible to WNV infection and a possible source for virus transmission [19], and the mortality of magpies due to lineage 2 WNV has been reported in Greece [20]. However, there are scarce data on the susceptibility of magpies to USUV infection. Free-living, apparently healthy, and actively collected magpies have been reported positive for USUV RNA in a region with co-circulation of USUV and WNV [21], and the seropositivity of magpies against USUV has been detected occasionally in large-scale serosurveys in France, Italy, and Germany [22,23,24,25,26]. To date, there is a lack of understanding of the interaction between USUV and WNV in avian hosts where the circulation of both viruses overlaps [11]. With this background, we experimentally addressed the susceptibility of magpies to USUV infection and their subsequent response to a lethal challenge with WNV. 

## 2. Materials and Methods

### 2.1. Experimental Design

Magpies were captured between April and June 2018 in hunting estates in south-central Spain under permit 346760/2018 of the regional government of the autonomic Community of Castilla-La Mancha (Spain), as part of a specific pest control program in the area. The presence of flavivirus (USUV and WNV)-specific neutralizing antibodies (NAbs) was tested using a plaque reduction neutralization test (PRNT) on Vero cells using twofold serial dilutions of heat-inactivated serum, as described in [27]. All birds were also tested for Flavivirus RNA in pooled oral and cloacal swabs and in their feather follicles [15]. 

A final group of 24 juvenile (less than one year old) magpies negative for USUV and WNV Nabs and RNA was transported to our biosafety level 3 (BSL-3) facilities, where they were housed in 2 separate boxes (Figure 1) in flight cages (12 birds/cage), equipped as described [19]. After one week of adaptation, animals were weighed and bled via the jugular vein for pre-inoculation serology. In the USUV box, one group of magpies (n = 9) was subcutaneously inoculated in the neck with 5 × 10^3^ plaque-forming units (pfu)/bird of USUV strain SAAR-1776 (GenBank accession no. KU760915.1 [28]), diluted in 200 µL Eagle Minimum Essential Medium (EMEM, BioWhittaker, Lonza, Verviers, Belgium), and 3 cage-mates were similarly sham-inoculated with medium alone and served as contact controls. On day 18 post-USUV infection, eight out of the nine USUV-infected birds were challenged with 5 × 10^3^ pfu/bird of WNV lineage 1 strain (GenBank accession no. KC407666; [29]). The remaining USUV-infected magpie and the three remaining USUV contact control birds were again sham-inoculated with medium alone. In the vehicle group (n = 12), housed in a separate cage (vehicle box), all magpies were inoculated with medium alone. On day 18, 9 birds were infected with WNV, and 3 were again inoculated with medium (contact control magpies) (Figure 1). Both viral inocula were back-titrated to confirm the injected dose. Food and water were provided *ad libitum* throughout the experiment. The magpies were monitored daily for clinical signs, and birds showing severe clinical signs were anesthetized with isoflurane and euthanized by intravenous injection of an overdose of sodium pentobarbital (Dolethal, Vetoquinol, Madrid, Spain), as were all surviving animals at the end of the experiment (37 days post infection (d.p.i.)). 

### 2.2. Sampling

At 0, 4, 7, 10, and 14 d.p.i. with USUV and at 4, 7, 10, 14, and 19 d.p.i. with WNV, all surviving animals were weighed and sampled (blood and feathers) as previously described [19]. Blood samples were allowed to clot overnight at 4 °C. Serum and all tissue and feather samples were stored at −80 °C until analysis. Growing feathers with pulp (vascular feathers) were collected from the dead and euthanized birds and placed into 0.5 mL of EMEM medium. Samples of brain, heart, and kidney were also collected into sterile containers and frozen at −80 °C until the analysis. In addition, cerebrum, cerebellum, medulla oblongata, heart, lung, spleen, liver, kidney, caecal tonsils, and duodenum samples were collected from deceased birds during a *post mortem* examination and were fixed in 10% neutral buffered formalin for histopathologic examination. 

### 2.3. Immunological and Viral Assays

Neutralizing antibodies for either virus (USUV and WNV) were analyzed using a plaque reduction neutralization test (PRNT) [30]. Titers were calculated as the reciprocal of the serum dilution, and diluted by at least 1:20, which reduced the plaque formation ≥90% (PRNT90). Sera were considered specific when only one of the viruses was neutralized or the titers were ≥4 times higher for one of them. Collected sera and vascular feathers were also tested for flavivirus (USUV and WNV) infectivity by plaque assay on Vero cell culture, as previously reported [31,32].

The presence of WNV RNA was analyzed in the serum, homogenized tissue, and processed feather samples, as previously described [19], and RT-qPCR was performed using specific primers of the USUV 3′ non-coding region [33].

### 2.4. Histopathology

Formalin-fixed tissue samples were trimmed, embedded in paraffin, and processed to obtain 4 µm sections that were stained with hematoxylin and eosin. These were independently examined by 2 investigators (U.H. and MC.C.) to determine the presence of USUV or WNV infection-associated lesions, respectively.

### 2.5. Immunohistochemistry (IHC)

Sections of the formalin-fixed paraffin-embedded tissue samples were routinely processed for IHC using the avidin–biotin–peroxidase complex (ABC) method described by Gamino et al. [15], with some modifications. Briefly, endogenous peroxidase activity was exhausted by incubation with 0.3% hydrogen peroxide in methanol for 30 min at room temperature (RT). The sections were incubated with 0.2% proteinase K (Sigma-Aldrich, St. Louis, MO, USA) in 0.05 M Tris-buffered saline (TBS; pH 7.6) and treated in a microwave oven at 37 °C for 8 min for antigen retrieval; after pretreatment, the sections were covered with 20% normal goat serum (Vector Laboratories, Burlingame, CA, USA) in 0.01 M phosphate-buffered saline (PBS) at RT for 30 min. For WNV and USUV antigen detection, a polyclonal antibody against the envelope protein E (BioReliance, Product 81–015, Rockville, MD, USA) was used in a 1:1000 dilution at 4 °C overnight. Following this, the sections were incubated for 30 min at RT with biotinylated goat anti-rabbit IgG secondary Ab (Vector Laboratories, Burlingame, CA, USA) diluted 1:200 in TBS containing 10% normal goat serum. All tissue sections were finally treated with ABC complex (Vectastain ABC Elite Kit; Vector Laboratories Inc., Burlingame, CA, USA) for 1 h at RT, then rinsed in TBS and incubated in chromogen solution (NovaRED Substrate Kit; Vector Laboratories Inc., Burlingame, CA, USA). Finally, the slides were counterstained with Harris hematoxylin. Tissue sections of the magpies that tested positive for WNV by RT-qPCR served as positive controls. Negative controls included the substitution of the primary antibody by 10% normal goat serum and a negative rabbit antibody (product 81-015; BioReliance, Rockville, MD, USA), as well as tissue sections of non-infected (WNV/USUV RT-qPCR negative) magpies.

To evaluate the number of immunolabeled cells on tissue sections, cell counts were carried out by three observers (M.-C.C., M.A.R., and L.M.) in 20 randomly chosen fields, of 0.2 mm^2^ whenever possible, and they were blinded to the group that was being analyzed. The results were given as the number of positive cells per 0.2 mm^2^. Cellular identification was based on the morphologic features, location, and size of the cells.

### 2.6. Statistical Analyses

Statistical analyses were performed using Graph Pad Prism for Windows, version 6 (Graph Pad Software, Inc., San Diego, CA, USA, 2005). Kaplan–Meier survival curves were analyzed by a log-rank test. Two-way analysis of variance (ANOVA) with Bonferroni’s correction for multiple comparisons was used to compare the proportion of change in body weight of the animals in the two groups throughout the experiment. An unpaired t-test was used to compare viremia and immunolabeled cells between the groups infected with the two viruses used. Statistically significant differences are indicated by asterisks (*) (*p* < 0.05).

## 3. Results

None of the magpies inoculated with 5 × 10^3^ PFU of USUV died (Figure 2), showed apparent signs of disease, or showed significant weight loss (Figure 3). No viremia, USUV genome, or USUV neutralizing antibodies were detected in their sera at any time point post-USUV analysis, 4, 7, 10, and 14 days post-infection. Similarly, none of the feathers collected at the same time points had USUV RNA. 

Upon WNV challenge, a significantly higher survival rate (n = 6/8, 75%, *p* = 0.0218) was recorded in USUV-exposed magpies than in the unexposed vehicle birds (n = 2/9, 22%) (Figure 2). 

Notably, none of the surviving USUV-infected magpies challenged with WNV showed any signs of disease. In contrast, as previously reported [19], clinical signs of disease were observed in magpies infected with WNV alone that died, including lethargy, ruffled feathers, ataxia, inability to fly, and leg paralysis.

WNV viremia was detected 4 days post-WNV infection in 75% (6/8) of the previously USUV-exposed birds and 78% (7/9) of the vehicle birds, respectively (Figure 4A). A tendency for the mean viremia to be lower in previously USUV-exposed magpies was evident but was not statistically significant. Only one animal in each group was viremic 7 days post-WNV infection. Similarly, infectious WNV was detected in the vascular feathers of birds from both groups from 4 to 10 d.p.i. (Figure 4B). WNV-specific neutralizing antibodies were recorded from 4 d.p.i. in both groups of animals until the end of the experiment (Figure 4C). These antibodies also had a neutralizing capacity against USUV but with titers that were >4 times lower than those against WNV, indicating that neutralization was due to cross-reactivity (Appendix A).

No mortality was recorded in any of the seven contact magpies housed with WNV-challenged cage-mates, but four (three from the USUV and one from the vehicle group) were viremic 7 days after WNV infection of their cage-mates (Figure 5A), including the one previously infected with USUV. Two of them were still viremic three days later (day 10 p.i. of their cage-mates). Viral titers in these contact birds were lower and detected 3 days later than in their experimentally infected cage-mates. Infectious virus was also recovered from the vascular feathers of these magpies from 7 days post-challenge of their cage-mates, thus with a delay similar to that of viremia (Figure 5B). Contact-infected birds also developed specific WNV-NAbs after 10 d.p.i. of their cage-mates (Figure 5C). Thus, in all of the birds infected by contact, the variables analyzed (viremia, infectious virus in vascular feathers, and NAbs) showed a delay with respect to the experimentally infected magpies. 

Macroscopic lesions observed during *post mortem* examination of magpies that had died from WNV infection in either group included generalized congestion, especially marked in the brain; swollen kidneys; and enlarged liver and spleen. Microscopical lesions were more severe in the animals that died between 7 and 10 d.p.i., and very similar between the two previously USUV-exposed and the USUV-unexposed (vehicle) magpies, although slightly less extensive in the former. The most severely affected tissues were the spleen, liver, kidney, heart, brain, and intestine. In general, the main microscopic findings were the presence of vascular changes, such as congestion and hemorrhages, as well as inflammatory infiltrates, cellular degeneration, and necrosis. In the central nervous system, the most predominant lesions were vasculitis, gliosis, neuronal necrosis, and mild-moderate satellitosis, compatible with a moderate multifocal acute non-purulent encephalitis (Figure 6A). Lesions in the heart were dominated by an acute mild to moderate myocarditis characterized by the degeneration of myocardiocytes and a diffuse mild lymphohistiocytic infiltrate, as well as swelling of the endothelial cells. In the liver and kidneys, the most important lesions were an acute moderate multifocal hepatitis, acute multifocal interstitial and tubulo-nephritis characterized by a moderate multifocal mononuclear inflammatory infiltrate and associated multifocal coagulative necrosis in hepatocytes and renal tubular epithelial cells (Figure 6B–D). A brownish pigment (likely hemosiderin) was observed in the cytoplasm of the Kupffer cells in the liver and was also especially abundant in the spleen (Figure 6E). In addition, lymphocyte depletion, necrotic foci of lymphoid cells, and severe hemosiderosis were present in the spleen. In the duodenum, an acute mild enteritis was present that was characterized by a diffuse moderate lymphohistiocytic inflammatory infiltrate in the lamina propria and submucosa, leading to a thickening of the villi and mild to moderate crypt hyperplasia (Figure 6F).

The antigen distribution in the tissues studied was assessed by IHC, and the antigen labeling in USUV-exposed magpies that died after WNV infection appeared to be considerably less intense and restricted to significantly fewer tissues than in the vehicle magpies that had succumbed to WNV infection (Chi-square, df 50, 1, z = 7.071, *p* < 0.0001, Figure 7 and Figure 8). As an example, the WNV antigen was completely absent in the hearts and kidneys of USUV-exposed magpies, two of the organs highly positive in the vehicle birds (Figure 8). The duodenum was the only tissue in which WNV antigen staining was more intense in the previously USUV-exposed magpies, although the difference was not significant.

## 4. Discussion

Currently, at least two mosquito-borne flaviviruses, USUV and WNV, are co-circulating in Europe, accounting for avian, horse, and human outbreaks, including a worrisome human mortality (https://ecdc.europa.eu accessed 3 June 2021). Birds are the main hosts for both viruses, with over 60 families affected [11,34], and corvids are recognized as especially susceptible to WNV [19,35,36].

In this respect, the Eurasian magpie, one of the most abundant corvids in Europe [18], is highly susceptible to WNV infection and is a possible source for virus transmission [19]. It has also been postulated as a sentinel for WNV activity [23,37]. However, data on the susceptibility of birds to experimental infection with USUV are only available for domestic chicken, geese, and canaries [12,13,14]. Here, we aimed to investigate the susceptibility of magpies to an experimental USUV infection and the possible protective response elicited against a lethal challenge with WNV. 

We showed that under experimental conditions, USUV-exposed magpies presented a higher survival rate (75%) when challenged with a lethal WNV strain compared to those not exposed to USUV (22%). Cross-protection against WNV in mice previously infected with USUV [29,33] or Zika virus (ZIKV) [27], and a consequent boost of neutralizing antibodies, have been previously described. However, data on the role of cross-protection in birds consecutively infected by two different flaviviruses are scarce and debated [38,39]. This can be exemplified by the response induced in experimental infections with WNV and St. Louis encephalitis virus (SLEV), which suggested that the level of cross-protective immunity depends on the time elapsed between heterologous infections [40,41,42]. Surprisingly, our data show that reduced mortality is not related to an increase in protective antibodies, suggesting instead a potential involvement of the cellular immune response for the outcome observed during WNV infection, an aspect that needs further investigation. Another explanation includes the generation of non-neutralizing antibodies against the non-structural NS-1 protein (e.g., see [43]) as these would not be detected by our PRNT. Under field conditions, healthy magpies positive for USUV antibodies have been detected in low numbers, both in areas with and without WNV activity [21,22,23,24,25,26], but to the authors’ knowledge, fatal natural infections of magpies by USUV have not been documented. 

Our experimental setting did not allow for a comparative study of the pathogenesis of WNV infection in previously USUV- and vehicle-inoculated magpies (e.g., by euthanasia and study of individuals on days 3, 6, and 10 post-WNV challenge). However, as two of the USUV-inoculated magpies died, we were able to compare lesions and viral antigen distribution in terminal cases. As expected, the lesions caused by the challenge with WNV in fatally infected magpies were very similar and consistent with previously described WNV pathology [36,44,45]. In contrast, significantly less viral antigen was detected by IHC in most tissues of the previously USUV-inoculated magpies (Figure 7 and Figure 8), except the duodenum, which suggests less extensive viral replication. As we did not find higher titers of neutralizing antibodies, the mechanism underlying this feature remains to be elucidated. The higher number of antigen-positive cells in the duodenum of previously USUV-exposed magpies, although not significant, should also be further explored to determine if this is a consistent finding or if it is mediated by variation between individuals. In contrast to other tissues, we do not have data on the detection of infectious virus or WNV/USUV RNA in the duodenum. Thus, as this polyclonal antibody has previously been shown to have a high affinity with different *Flaviviruses,* including USUV [46], although unlikely, we cannot completely rule out that part of the positive signal in this location is from USUV replication. Further additional studies should elucidate this in the future.

The detection of infective virus and WNV-specific antibodies in 4 out of 7 contact magpies housed with the WNV-challenged ones, including the one exposed to USUV, confirms, as recently suggested [19], that WNV infection by contact occurs in magpies. Importantly, prior exposure to USUV does not seem to prevent transmission, likely due to the fact that despite inducing cross protection, it does not reduce the replication of the virus in blood and vascular feathers, and, although not tested in our study, likely not in cloacal excretion either. Thus, our experimental results provide some evidence that supports the contribution of cross-protection of an abundant avian host against WNV infection by previous exposure to co-circulating USUV to the temporal scale of WNV epidemiology.

We were unable to establish a detectable USUV infection, since the magpies did not show clinical signs, elicited either NAbs or viremia, and no infectious virus nor viral RNA could be recovered from their vascular feathers after USUV infection. To some extent these results are in accordance with those previously reported in domestic chickens and geese, in which viremia, antibodies, and viral shedding were only sporadically observed [12,13]). In this respect, in Eurasian coots (*Fulica atra*), the immune response against USUV has been shown to be less robust and prolonged compared to that against WNV, for which specific neutralizing antibodies titers are higher [47]. In the field, live healthy magpies collected for active surveillance purposes were shown, by RT-PCR, to have USUV in their tissues, although the presence of infective virus was not tested, and the number of positive birds was very low (1/600 [23] and 2/399 [21]).

Additionally, an influence of age on susceptibility has previously been observed with other related flaviviruses in several avian species, such as in red-legged partridges (*Alectoris rufa*) [32], in chukar partridges (*Alectoris chukar*) [48], as well as in domestic geese [44,49]. In fact, the limited susceptibility to experimental USUV infection in the domestic birds mentioned above [12,13] was observed in two-week-old birds. This age-dependent susceptibility may be the reason for the lack of evidence of USUV infection as the magpies used in this study were several months of age. However, we cannot exclude the possibility that a short viremia window, cleared before 4 d.p.i., was missed, or even that of an abortive infection. Although this is unlikely, it would explain the absence of magpie mortality in USUV outbreaks in northern and central Europe [26]. Another possibility would be that a higher infectious dose of USUV is necessary in magpies (or wild birds in general). While we used 5 × 10^3^ pfu/bird in the experimental USUV infection, a recent study with red-legged partridges for USUV neutralizing antibody production employed a higher dose of infection (10^4^ TCID, potentially equivalent to approximately 5 × 10^4^ PFU) [49]. Finally, experimentally infected 10-month-old canaries (*Serinus canaris*) showed signs of disease and mortality, had USUV RNA in their serum from day one post-infection until the end of the experiment, and developed neutralizing antibodies against USUV, suggesting species-specific USUV susceptibility [14].

In conclusion, magpies survived experimental exposure to USUV, and the previous exposure of magpies to USUV greatly reduced mortality due to WNV. However, protection was not sterile since it did not prevent WNV infection, virus replication, or viremia. These results are highly relevant for flavivirus ecology and contention.

## Figures and Tables

**Figure 1 viruses-13-01409-f001:**
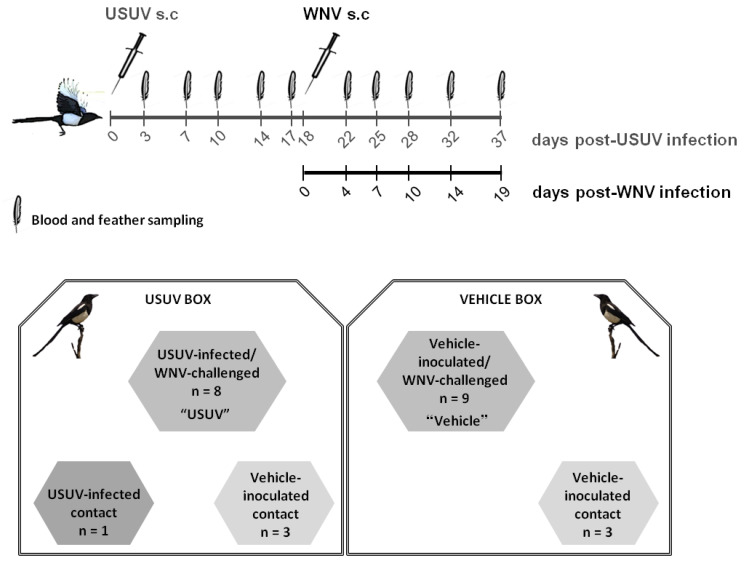
Experimental design. Representation of the USUV and WNV infection, sampling schedule, and distribution of the magpies in the experimental boxes.

**Figure 2 viruses-13-01409-f002:**
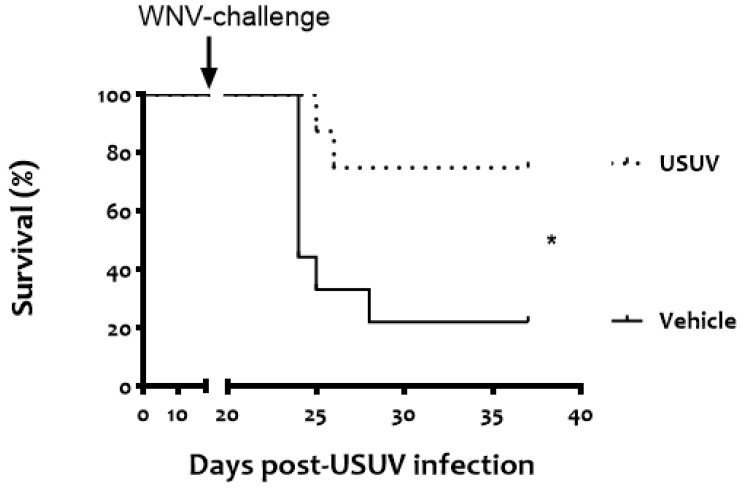
USUV exposure conferred protection against lethal WNV challenge. Survival rates in magpies infected with USUV (5 × 10^3^ pfu/bird) or sham-inoculated (vehicle) and challenged 18 d.p.i. with 5 × 10^3^ pfu/bird of WNV NY99. The asterisk represents a statistically significant difference between both groups (* *p* < 0.05).

**Figure 3 viruses-13-01409-f003:**
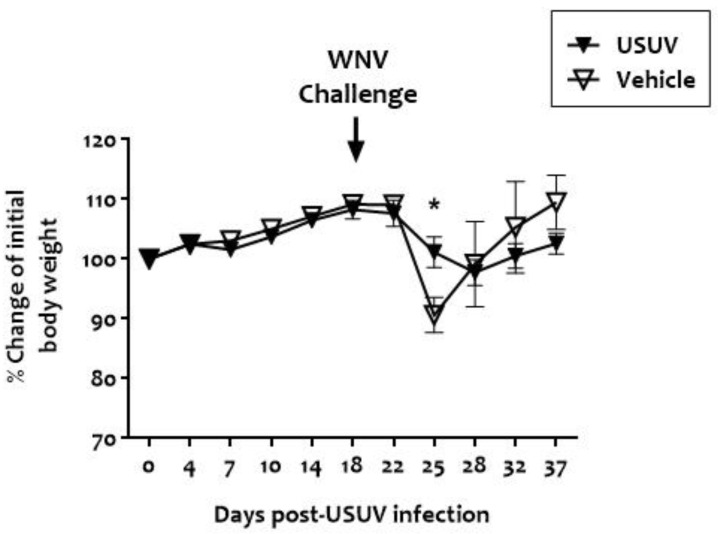
Proportion of change in body weight over time of sham-inoculated and USUV-infected magpies challenged with WNV. Significant differences are marked with an asterisk.

**Figure 4 viruses-13-01409-f004:**
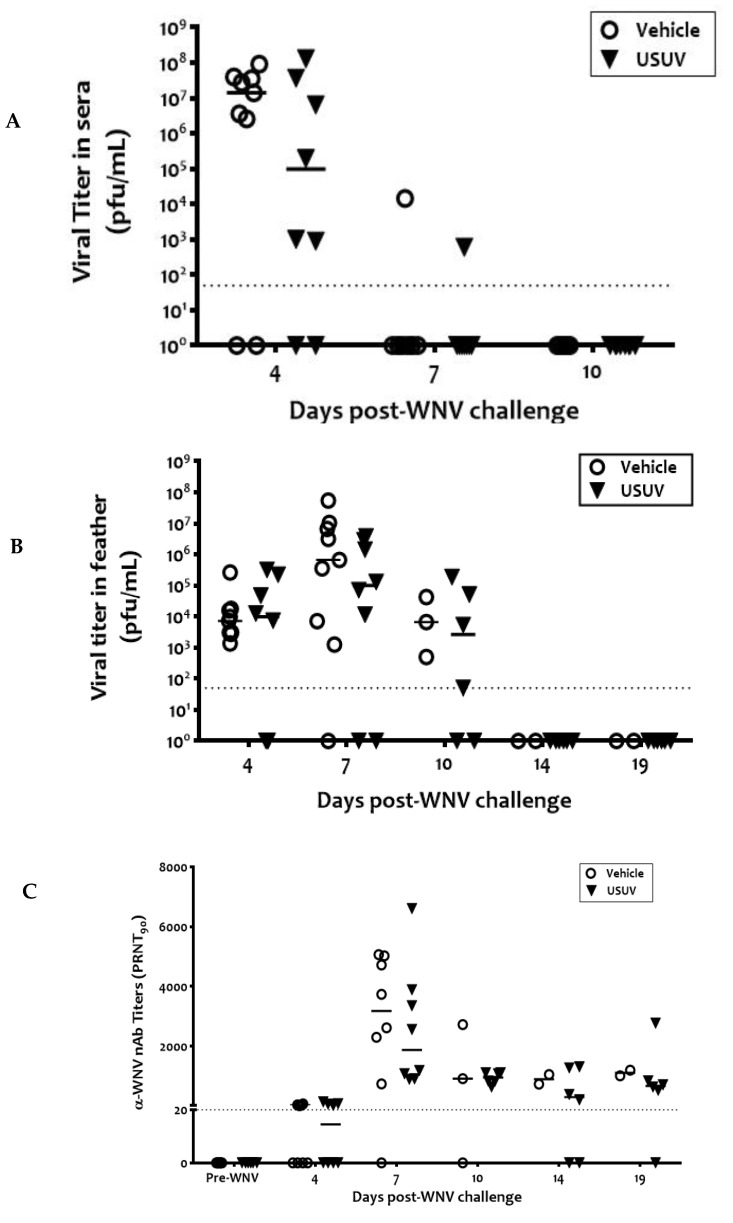
Infectious virus in sera and vascular feathers and neutralizing antibodies, developed by experimentally WNV-infected magpies. Infectious virus titers in sera (**A**) and vascular feathers (**B**) and neutralizing antibodies (**C**). Circles and triangles represent sham-inoculated vehicle birds and USUV-infected birds, respectively. Dotted lines represent the limit of detection of the assays.

**Figure 5 viruses-13-01409-f005:**
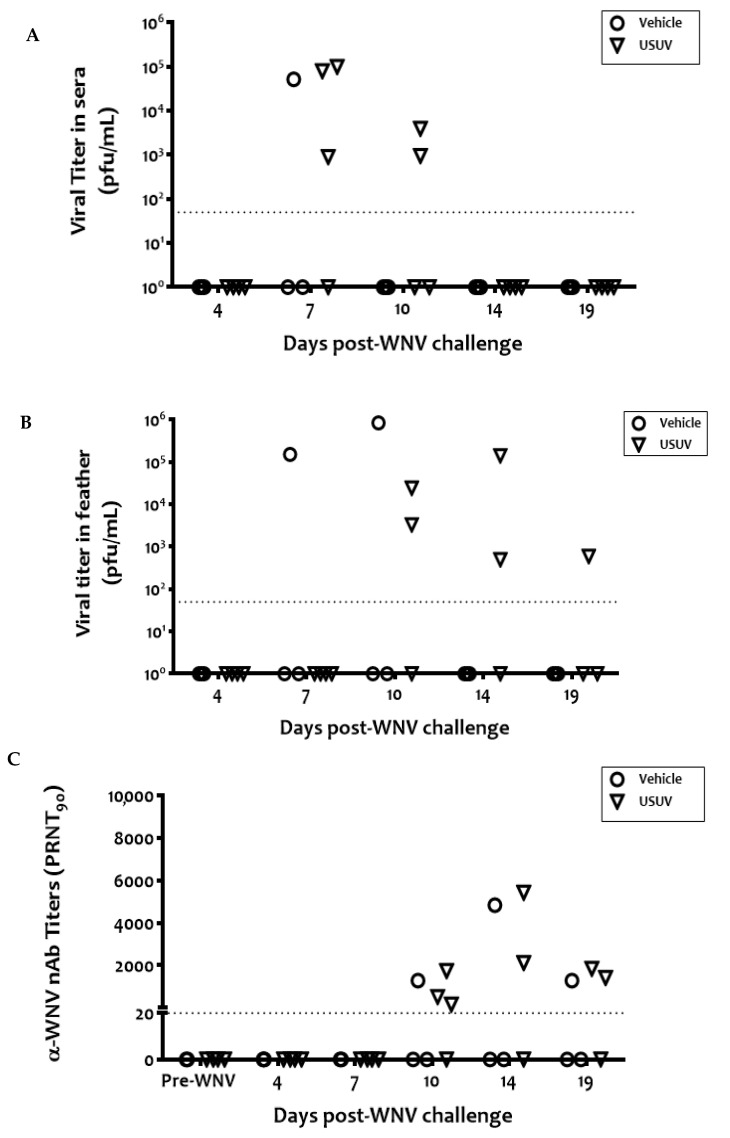
WNV contact transmission. Infectious virus titers in sera (**A**) and vascular feathers (**B**), and neutralizing antibody titers (**C**) in WNV contact-infected magpies. Circles and triangles represent animals housed with sham-inoculated vehicle birds and USUV-infected birds, respectively. Dotted lines represent the limit of detection of the assays.

**Figure 6 viruses-13-01409-f006:**
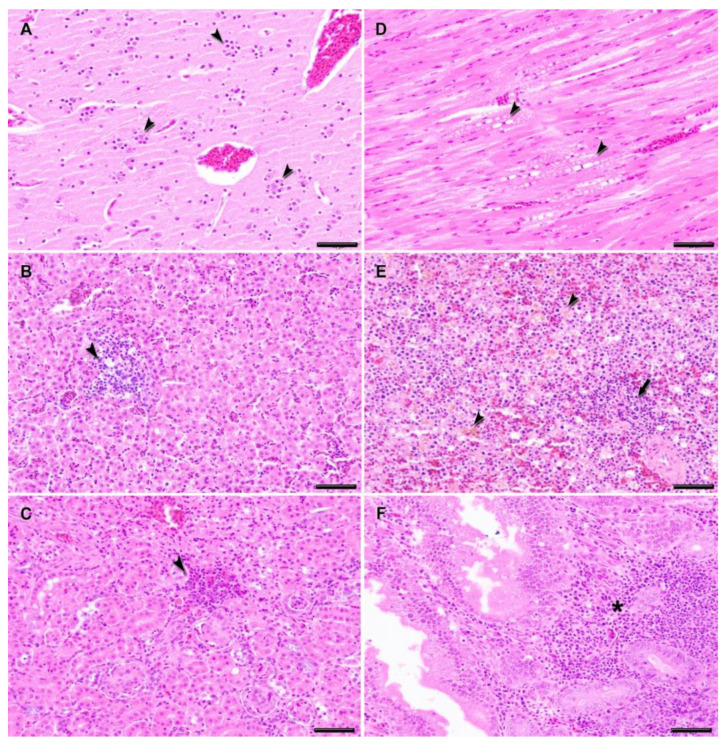
Histopathologic lesions in magpies experimentally infected with West Nile virus. (**A**) Diffuse moderate gliosis and satellitosis (arrowheads) in cerebrum. Multifocal mononuclear inflammatory infiltrate in the liver (**B**) and kidney (**C**) (arrowheads). (**D**) Focally extensive degeneration and necrosis of myocardiocytes (arrowheads). (**E**) Massive hemosiderosis (arrowheads) and lymphocytic necrosis (arrow) in spleen. (**F**) Mononuclear inflammatory infiltrate (asterisk) in lamina propria of duodenum. Hematoxylin and eosin staining; bars = 50 μm.

**Figure 7 viruses-13-01409-f007:**
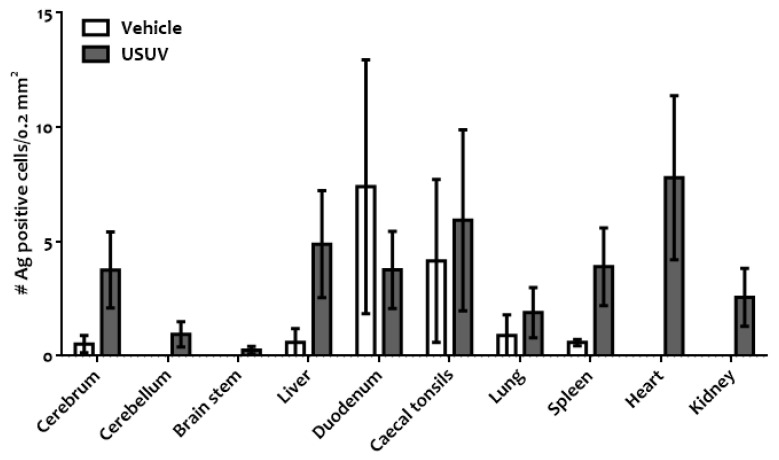
Viral antigen in cells of different tissues of fatally WNV-infected magpies. No. of antigen-positive cells/0.2 mm^2^ tissue section detected by immunohistochemistry using a polyclonal WNV antibody, in infected magpies from the vehicle group (n = 7) and the previously USUV-exposed, group (n = 2) that succumbed to the infection.

**Figure 8 viruses-13-01409-f008:**
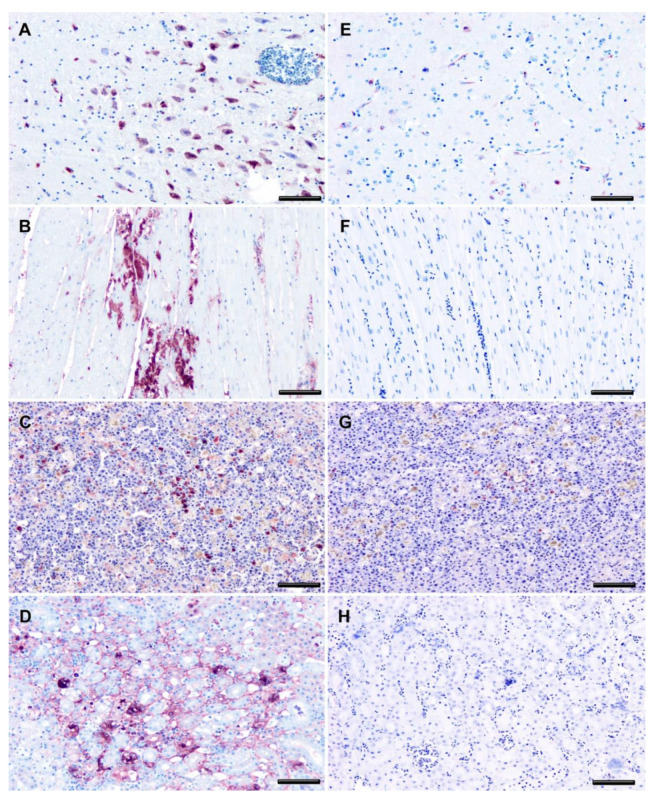
West Nile virus antigen distribution. Representative images of WNV-Ag distribution in the brain, heart, spleen, and kidney of infected magpies from vehicle group (**A**–**D**) and previously USUV-exposed group (**E**–**H**). Immunohistochemistry; bars = 50 μm. Staining intensity is reduced in previously USUV-exposed magpies.

## Data Availability

The data presented in this study are contained in the manuscript and additionally available on request from the corresponding author.

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
