# Peer review of "Previous Usutu Virus Exposure Partially Protects Magpies (Pica pica) against West Nile Virus Disease But Does Not Prevent Horizontal Transmission"

_viruses, 2021, doi:10.3390/v13071409_

Round 1
Reviewer 1 Report
see the annex

Author Response
The manuscript entitled “Previous Usutu virus exposure partially protects magpies (Pica pica) against West Nile virus but does not prevent horizontal transmission” by Escribano-Romero et al. provides first interesting insights into coinfections with USUV and WNV in birds. The manuscript is well written and demonstrates a lot of interesting results. However, a few major revisions must be completed before publication. The layout of the experiments is a bit confusing and is not conveyed in a clear manner. The discussion focuses mainly on USUV and is missing an analysis of the pathogenesis of WNV with and without USUV pre-infection. Please find below all my recommendations:
General recommendations:
- Why did you not have any environmental controls in your experiments? How can you asses
the effect of stress during the handling of the wild birds on your results without having any
possibility of comparison.
We completely agree that not having true environmental controls is a limitation. In this case unfortunately, both logistic reasons (boxes available) and animal welfare (reduction of animals used in experimentation) were the reasons. In a previous experiment using the same WNV strain as the challenge virus employed here an entire environmental control group under the same conditions in a different box, was included and maintained, handled and sampled the same way as in this study and did not show weight loss or mortality as compared to birds in the experimental groups (see doi: 10.1371/journal.pntd.0006394). Thus, despite the lack of concurrent environmental controls, we are fairly convinced that our results are representative of true infection-related effects. Having said this, it is evident that stress from handling and captivity but also in turn ad libitum access to food and water and the constantly controlled climate will always limit the degree with which results of experimental infections are representative of field situations, and thus results must be interpreted in this context.
- It also appears quite difficult to assess a monoinfection with USUV in magpies (antibody
production etc) on the basis of only one bird or was this not the purpose of the one sham inoculated USUV bird?
Monoinfection with USUV was evaluated in birds infected only with USUV (n = 8) prior to the boost with WNV, that is, for 18 days. The aim of having a single "USUV-infected contact" bird was to have an idea of whether previous USUV infection could reduce the already described WNV contact infection (10.3389/fmicb.2019.01133), although we are perfectly aware that a single bird does not yield any conclusive results.
- Please try and better explain the experimental set-up. The boxes in Figure 1 make it more
confusing rather than helping to explain the different groups. For example, in the text you do
not mention a vehicle-inoculated group which you then, however, refer to in the figure. I
believe the boxes are unnecessary, it would probably be better to focus on a good
explanation in the text.
Thank You. According to your suggestions we have amended the text and figure 1 that we still think of interest to maintain as for some readers a visual layout is clearer.
- The text does not clearly define at which timepoints the birds were euthanized/died.
In this study our focus was on disease and mortality as well as viremia and due to the number of birds in the study and the high mortality in the vehicle group no planned euthanasia were included in the study. Only diseased birds that showed clinical signs incompatible with life were euthanized. Timepoints at which birds died or were euthanized are shown in Fig. 2.
- Please by consistent in whether you capitalize P or not and in the use of the oxford comma
(eg. Line 59).
As suggested by reviewer 1, corrections have been done.
- The manuscript places a lot of focus on pathological findings yet they are not all mentioned
in the discussion, please discuss also the macroscopic/microscopic findings. Also, only USUV
infections in magpies are discussed yet not the results of the WNV infections.
Thank You for this remark. We have now included a paragraph discussing our pathological findings and the results on viral antigen detection in the tissues of deceased or endpoint euthanized magpies (lines 300-317)
As to the second part of Your comment, experimental WNV infection in magpies have already been described (10.3389/fmicb.2019.01133; 10.1371/journal.pntd.0006394) and, thus, its analysis was not an objective of this study.
- Please also discuss why you possibly could not detect any USUV-specific antibodies in all your
birds, since they must have been present to partially “protect” them against a WNV
infection.
As we discuss in the original version and now in the revised one (lines 291-295), we hypothesize that protection could be due to the involvement of a potent cellular immune response, more than to the presence of neutralizing antibodies and, that, in any case, this aspect needs further investigations. Also, non-neutralizing antibodies e.g. directed against the non-structural NS1-protein could be involved in the response, and these would not be detected by PRNT (e.g. see in Lim et al., 2012: http://dx.doi.org/10.4167/jbv.2012.42.2.108). We have included an additional phrase stating this possibility and included Lim et al. 2012 in the reference list.
Specific recommendations:
Line 48: I believe you are missing a word in this sentence: “During the last ten years…”.
Corrected
Line 53: The sentence is a missing an “a”: “…susceptible to a WNV infection”.
Corrected
Line 54: The sentence from Line 52 to 55 appears very long, I would recommend to split it:
“…transmission [19]. Mortality of…”
Corrected
Line 57: Please change the formatting of the reference 21: “]” rather than “}”
Corrected
Line 77: Please elaborate why you used this specific titer for the infection experiments (5x103
PFU/bird). Usually, higher titers are used to correspond with the virus titers often transmitted by vector competent mosquitoes.
We agree with the reviewer and, although the titers transmitted by mosquitoes are very variable depending of the species, frequently higher titers are used for experimental infections.
However, a titer of 103 -104 PFU/animal has been frequently successfully used by us and other authors for USUV and WNV experimental infections in animal models, including magpies (For example https://doi.org/10.1186/s12974-020-02060-4, http://dx.doi.org/10.1016/j.virol.2015.03.020), https://doi.org/10.3389/fmicb.2019.01133), and as we were more interested in the interaction of subsequent infections than in producing USUV-infection related mortality we chose to stay in the lower range of infectious dose.
Line 77: Please explain why you used the SAAR-1776 USUV strain from Africa and not a wellestablished European or even Spanish strain. The same is true for WNV, why did you decide to use an American strain, which is described to be more pathogenic than European ones? What cell passage number was used?
Both USUV and WNV strains are well characterized ones and have been used in our laboratory for many years, as their genomes and behaviors are very similar to the ones currently circulating in Europe (https://doi.org/10.1101/2020.08.31.275149).
Viral stocks have 3-5 cell culture passages.
Line 82: Reference 29-> is this really the correct reference for the specific isolate?
Yes, it is. It corresponds to the cell culture passaged strain that was fully sequenced and used in this study.
Line 82-84: Were the “control” (i.e., sham-inoculated) birds (monoinfected) kept in a separate cage and the coinfected birds and contact-birds kept in another cage? This is also not clear from the text.
The text has been modified in order to make the experimental setup clearer. The sham -inoculated (mono-infected) and co-infected birds were held in two different BSL-3 room (here called box), thus completely isolated from each other, and each of the groups had their contact-birds that were housed in the same flight cage in the same box (direct contact in order to mimick contacts in communal roosts, during feeding and drinking). Each bird was identified individually with a numbered metal ring which enabled the described etup.
Figure 1.: Why did you decide to take the first blood samples 4 dpi and not earlier eg. 3 dpi, often the time point of viremia?
Viremia titers peak in magpies between 3 to 4 d.pi. and titers from 4dpi onwards are high enough to be detected based on our previous experience (10.1371/journal.pntd.0006394;
10.3389/fmicb.2019.01133). In addition, in this way samplig was conducted a regular interval.
Line 94: Do you think leaving the blood samples over night at room temperature might have
influenced your PCR/PRNT results?
We regret that this information was missed in the original manuscript. Samples were kept at 4 ºC overnight, not at RT. This has now been corrected into the revised version
Line 98: brain= cerebrum
Corrected
Line 128: Does this antibody efficiently detect both viral antigens? If yes, why did the authors not include an Usutu positive control in their examination? Without it, an exclusion of Usutu antigen in the first group is not possible.
Thank You for this comment. Yes, in fact this antibody very efficiently detects Flavivirus antigen, and we have used it previously for detection of WNV (doi: 10.3390/pathogens10060748), USUV (doi: 10.3201/eid1907.130199) and BAGV (doi: 10.1186/1297-9716-43-65). You are right that due to the cross-reactivity a signal could also be from USUV antigen. In this case there are several reasons for which we are confident that the observed signal indeed was form West Nile virus. On one hand the collected tissues (kidney, heart, brain) when tested by Realtime RTPCR gave clear positive results for WNV but not USUV. On the other hand, we included an USUV positive control (tissue from magpies only infected with USUV and euthanised on D3 after infection, data not included) in the study. However, we do not have any Real time RTPCR results nor IHC data on the duodenum, and although we think it highly unlikely, here we cannot completely rule out that the more intense staining could be due to local persistent replication of USUV. We have included a comment on this fact in the discussion (lines 308ff.).
Line 171, 200: It remains unclear how many animals died from the WNV infection and at which time point. It would be helpful for the reader to have a table summarizing the most important results at least of the clinical and immunohistochemical results.
Although we thank You for Your comment, we believe that data in the text and figure 2 offer enough information in this regard, and that not additional table is needed. However, we can add it, either in the text or a supplementary table, if the Editor considers it necessary.
Figure 2: Please use a better term than “control”- it is not clear which you mean, the contact birds or the single bird monoinfected with WNV? The asterisk should not be written in brackets (as in the next figure) and should be positioned between the USUV+WNV infected and the “controls”.
Thank You, according to Your suggestions the term "control" has been replaced by "vehicle" in the figures and text.
Line 181: “…4 were viremic…” -> 4 from which contact group now?
By referring to the mentioned figure 5a (WNV contact transmission), the reader can see that 3 of them belong to the USUV box and one of them to the vehicle one. However, this has now been clarified in the revised version (lines 76ff and line 201).
Figure 4: 0 does not exist in a logarithmic scale, please correct. The text in figure 4c appears much smaller than in the other two figures.
The difference in font size is due to the reformatting done when transferring to Word, since it is a longer figure. In the original format it has the same font size. On the other hand, the scale on figure 4C, contrary to 4A and 4B, is not logarithmic since we are measuring antibody titers, not viral ones. We have corrected the figures where appropriate. However the corrected compositions of figures 4a-c and 5a-c is submitted separately as it could not be included in the manuscript without mixing up the page and line counts completely
Line 199: “post morten” in italics? Like was done in line 100.
Corrected
Figure 6: (F) you are missing a reference to the star in the figure in your caption.
Corrected. Thank You
Lines 201-215: Correct histopathological diagnosis are mostly missing, this is just a (nice) description of pathomorphological alterations, but the interpretation is mostly up to the reader. For example, the infiltration in the gut, is this just an infiltration as it is normally seen or is it already an Enteritis. The same is true for the heart, what kind of inflammation is it? Necrotizing or just lymphohistiocytic. A full pathological diagnosis includes localisation (diffuse, multifocal, focal), degree (mild, moderate, severe), the time (acute, subacute, chronic), the character (suppurative, non-suppurative or lymphohistiocytic etc.) and the organ/tissue (a vasculitis or an encephalitis, myocarditis…).
We apologize for the incomplete description/diagnosis. We have now expanded the results section on pathological lesions (lines 227ff).
Line 242: Please split the sentence into two: “In this respect, the Eurasian magpie, one of the most abundant corvids in Europe [18], is highly susceptible to a WNV infection and a possible source for virus transmission [19]. It has also been postulated as a sentinel for WNV activity [37, 23].”
Corrected
Line 246: As far as I am concerned the comma in front of “and” is not necessary: “Here, we aimed to investigate the susceptibility of magpies to an experimental USUV infection and the possible protective response elicited against a lethal challenge with WNV.”
Corrected
Line 256: “[” rather than “{”
Corrected
Line 271: “Importantly, prior exposure to the USUV does not seem to prevent transmission, likely because, despite inducing cross protection, it does not reduce replication of the virus in blood and vascular feathers, and, although not tested in our study, likely neither cloacal excretion.” How come viral excretion was not included in this study?
Thank You for this comment. At the time of the design of this experiment, and as we are dealing with wild birds, and as in our previous studies (10.1371/journal.pntd.0006394) feathers were as informative as cloacal swabs, to reduce animal manipulation and stress we decided to only collect feathers. Our primary focus was on the influence of USUV exposure on the degree of development of WNV viremia thus unfortunately we lack data on cloacal/oral excretion.
Line 288: “(1/600, [23] and 2/399, [21].” You are missing the closing bracket and I also believe the commas are unnecessary here.
Corrected
Line 293: “This age-dependent susceptibility could be behind of the lack of evidence of USUV
infection as the magpies herein used were several months of age.” Please rephrase – “could be
behind” does not sound correct. Also, please already state in the material and methods section that the birds were several months old / < 1 year.
Correction added to materials and methods section and sentence rephrased.
Line 295: “However, we cannot exclude that a short viremia window, cleared before 4 d.p.i., was missed, or even an abortive infection.” Please elaborate, before beginning your experiments did you only check for flavivirus-specific antibodies or did you also check for virus RNA via PCR?
The sentence mentioned by reviewer 1 refers to the experimental phase, i.e. we mean that by sampling (as You also remarked), on day 4p.i. we may have missed an extremely short abortive USUV infection in our experimental birds.
Regarding the analysis prior to the start of the experiment, each bird was tested twice by PCR on feather follicles for viral RNA and for Flavivirus specific antibodies prior to transport to the BSL-3 facility. During this period the magpies were kept in mosquito net – fitted mostly indoor -flying cages.
Line 300: “While we used 5x103 pfu/bird in the experimental USUV infection, a recent study with red-legged partridges (Alectoris rufa) for USUV neutralizing antibodies production employed a higher dose of infection (104 TCID, potentially equivalent to approximately 5x104) [47].” Please state the latin name for the birds the first time they are mentioned and not at a later time point. “… equivalent to 5x104” –> do you mean PFU?
Corrected
Line 302: “Finally, experimentally infected10-month-old canaries…” you are missing a space between “infected” and “10”.
Corrected
Lines 306-309: The conclusion is very short and should place more focus on the relevance of your results. The last sentence is identical to one in the abstract.
We think this is a concise conclusion and do not believe that there is much room to expand it.
The sentence is the same because it is the key conclusion of this study and we want to showcase it in the abstract.
Lines 319-329: In the Institutional Review Board Statement you mention a lot of facts which also belong in the materials and methods section as for example when the birds were euthanized because they reached their human endpoint or because the experiment was terminated. Why do you suppose so many birds died due to the infection?
We have now also included the signposted information in the M&M section of the manuscript.
Magpies (corvid family) are highly susceptible to WNV infection (10.1371/journal.pntd.0006394;
10.3389/fmicb.2019.01133) and high mortality has also been described in a wnv lineage two outbreak in the natural environment (doi: 10.3201/eid2512.181225). Also, clinical signs and virological and immunological results support that the magpies died of WNV infection.
Reviewer 2 Report
The manuscript by Escribano-Romero et al is an interesting report of cross-protection between Usutu virus and West Nile virus in birds. The authors evaluate USUV in magpies, which do not develop viremia or disease, followed by WNV challenge. Birds previously challenged with USUV are protected against morbidity and mortality caused by WNV and sustain lower viremias (though this not statistically significant). Surprisingly, the USUV-exposed birds do not develop cross-neutralizing responses to WNV. Some additional controls for the PRNTs would be helpful – ie, do the birds develop neutralizing responses against USUV after WNV challenge?
Major comments:
- Did any of the birds develop USUV neutralizing responses? The way the manuscript is written, it’s unclear whether USUV PRNTs were performed—could this data be added to Figs 4C and 5C, even if the data is negative?
- Lines 261-265 – in order to evaluate a boost in neutralizing antibodies, the authors need to be measuring responses against the primary virus (Usutu virus), both before and after WNV challenge. It’s unclear if this was done.
- Add the word “disease” to the title – Previous Usutu virus exposure partially protects magpies (Pica pica) against West Nile virus disease but does not prevent horizontal transmission
Minor comments:
- Add limit of detection to Figs 4 and 5
- What type of y-axis is used for the PRNT data? It doesn’t look linear.
- Line 241 – add “to WNV” at the end of the sentence
Author Response
Major comments:
- Did any of the birds develop USUV neutralizing responses? The way the manuscript is written, it’s unclear whether USUV PRNTs were performed—could this data be added to Figs 4C and 5C, even if the data is negative?
PRNT analyses were performed to detect antibodies against USUV (line 109f). As indicated in the original text and now in the revised version (lines 160ff), the pre-WNV challenge analysis did not detect specific neutralizing response against USUV. After the WNV challenge, antibodies with neutralizing capacity against USUV were found in both vehicle and USUV infected magpies (Supplementary Figure 1). However, titres were <4 times lower than against WNV, which is also an indication that they were raised against WNV although presenting some cross-reactivity against USUV. These results clearly support that these are antibodies developed against WNV that show cross-reactivity for USUV. This is a very well-known characteristic of mosquito-borne flaviviruses.
- Lines 261-265 – in order to evaluate a boost in neutralizing antibodies, the authors need to be measuring responses against the primary virus (Usutu virus), both before and after WNV challenge. It’s unclear if this was done.
In line with your previous comment and, as it can be seen in both figure 4C and S1, there was no boost of antibody response in either USUV or vehicle groups after WNV challenge. The data show the usual kinetics of antibody response that appears in magpies infected with WNV (10.1371/journal.pntd.0006394; 10.3389/fmicb.2019.01133), suggesting again that USUV is not inducing a proper neutralizing antibody response.
- Add the word “disease” to the title – Previous Usutu virus exposure partially protects magpies (Pica pica) against West Nile virus disease but does not prevent horizontal transmission
Correction has been done.
Minor comments:
- Add limit of detection to Figs 4 and 5
Modification has been done.
- What type of y-axis is used for the PRNT data? It doesn’t look linear.
The figure was done with Graph Pad Prism programme and the y-axis scale for the PRNT data is linear divided into two segments and the format number is decimal
- Line 241 – add “to WNV” at the end of the sentence
Correction has been added.
Round 2
Reviewer 1 Report
The modifications and corrections added by the authors suit me good.
Only some minor revision are necessary :
Line 230-236: The pathological findings are much better now with the complete diagnosis. However, this has led to a long sentence from Line 230 to 236. I would recommend splitting it. There are also two commas in line 234.
Line 301: I believe you are missing a “with”: “…caused by the challenge with WNV”
Line 304: For better comprehension, please add, “of previously USUV inoculated magpies” after “significantly less viral antigen was detected by IHC in most tissues”. If that is what you wanted to express.
Line 311: There is a space missing in “to have a high affinity”.
Author Response
Thank You very much for Your comments. We have made all suggested corrections in the text.